# Structural basis for assembly and function of a diatom photosystem I-light-harvesting supercomplex

Ryo Nagao[1,11], Koji Kato[1,11], Kentaro Ifuku [2], Takehiro Suzuki[3], Minoru Kumazawa[4], Ikuo Uchiyama[5], Yasuhiro Kashino [6], Naoshi Dohmae[3], Seiji Akimoto [7], Jian-Ren Shen [1], Naoyuki Miyazaki [8,9✉] & Fusamichi Akita [1,10✉]

Photosynthetic light-harvesting complexes (LHCs) play a pivotal role in collecting solar energy for photochemical reactions in photosynthesis. One of the major LHCs are fucoxanthin chlorophyll $a/c$-binding proteins (FCPs) present in diatoms, a group of organisms having important contribution to the global carbon cycle. Here, we report a 2.40-Å resolution structure of the diatom photosystem I (PSI)-FCPI supercomplex by cryo-electron microscopy. The supercomplex is composed of 16 different FCPI subunits surrounding a monomeric PSI core. Each FCPI subunit showed different protein structures with different pigment contents and binding sites, and they form a complicated pigment–protein network together with the PSI core to harvest and transfer the light energy efficiently. In addition, two unique, previously unidentified subunits were found in the PSI core. The structure provides numerous insights into not only the light-harvesting strategy in diatom PSI-FCPI but also evolutionary dynamics of light harvesters among oxyphototrophs.

[1] Research Institute for Interdisciplinary Science and Graduate School of Natural Science and Technology, Okayama University, Okayama 700-8530, Japan. [2] Graduate School of Biostudies, Kyoto University, Kyoto 606-8502, Japan. [3] Biomolecular Characterization Unit, RIKEN Center for Sustainable Resource Science, Saitama 351-0198, Japan. [4] Faculty of Agriculture, Kyoto University, Kyoto 606-8502, Japan. [5] National Institute for Basic Biology, National Institutes of Natural Sciences, Aichi 444-8585, Japan. [6] Graduate School of Life Science, University of Hyogo, Hyogo 678-1297, Japan. [7] Graduate School of Science, Kobe University, Hyogo 657-8501, Japan. [8] Institute for Protein Research, Osaka University, Osaka 565-0871, Japan. [9] Life Science Center for Survival Dynamics, Tsukuba Advanced Research Alliance (TARA), University of Tsukuba, Ibaraki 305-8577, Japan. [10] Japan Science and Technology Agency, PRESTO, Saitama 332-0012, Japan. [11] These authors contributed equally: Ryo Nagao, Koji Kato. ✉email: naomiyazaki@tara.tsukuba.ac.jp; fusamichi_a@okayama-u.ac.jp

Oxygenic photosynthesis is a fundamental biological process with which solar energy is converted into biologically useful chemical energy and dioxygen is formed, thereby sustaining almost all life forms on the earth[1]. The photosynthetic light-energy conversion reactions are performed by two multi-subunit membrane protein complexes, photosystem I (PSI) and II (PSII). To harvest enough energy for the photochemical reactions, photosynthetic organisms have developed various light-harvesting pigment–protein complexes (LHCs), which play crucial roles in supplying energy to the photosystems by capturing unique spectral components of the light. This is achieved by different types and numbers of pigments including chlorophylls (Chls) and carotenoids (Cars) that are bound to LHCs, resulting in variations in the color of the organisms that we see in our life[2]. Based on the color variations, oxyphototrophs have been divided mainly into green and red lineages owing to the different types of pigments bound to LHCs, namely, the green lineage has Chl $a/b$-binding LHCs, whereas the red lineage has Chl $a$ or Chl $a/c$-binding LHCs, respectively. These linages also have different Car compositions. These differences are important for oxyphototrophs to survive under different light conditions.

In order to elucidate the mechanisms of light-energy capture and transfer within the photosystem-LHC supercomplexes, their structural information is indispensable and will also provide insights into why the color variations occur in the oxyphototrophs. The structures of green lineage PSI-LHCI supercomplexes have been determined from higher plants by X-ray crystallography[3,4] and from green algae by cryo-electron microscopy (cryo-EM)[5–8]. These structural analyses revealed the complicated pigment–protein networks in the PSI-LHCI, offering a structural basis for the energy flow from LHCI to PSI in the green lineages. Moreover, they revealed evolutionary diversity of LHCI from green algae to higher plants. In the red lineages, on the other hand, the PSI-LHCI structure was solved from a primitive red alga, *Cyanidioschyzon merolae* by cryo-EM[9] and X-ray crystallography[10], which showed the presence of two types of PSI-LHCI, one with three Lhcr subunits and the other one with five Lhcr subunits. However, the organization and structure of PSI-LHCI from the red-lineage oxyphototrophs via a secondary endosymbiosis event of red algae are still missing. Among the red lineages, diatoms are one of the major clades in oxyphototrophs and possess unique LHCs that bind fucoxanthin (Fx) and Chl $a/c$, which are called Fx Chl $a/c$-binding proteins (FCPs). We have purified and characterized a supercomplex consisting of PSI with specific FCPs (PSI-FCPI) from a marine centric diatom *Chaetoceros gracilis*, and analyzed its excitation energy transfer (EET) properties[11–15].

Here, we report a 2.40-Å resolution cryo-EM structure of the PSI-FCPI supercomplex isolated from *C. gracilis*. The structure revealed the presence of 16 FCPI subunits associated with the PSI core, and identified most of the pigments they bind, including Chls $a$ and $c$, Fx and diadinoxanthin (Ddx) clearly, thereby revealing a highly complicated pigment–protein network involved in EET and energy quenching.

## Results

**Overall structure of the PSI-FCPI supercomplex.** The cryo-EM images of the PSI-FCPI were obtained by a Titan Krios electron microscope. After data processing of the resultant images by RELION (Supplementary Fig. 1 and Supplementary Table 1), a final density map was obtained for the diatom PSI-FCPI with a C1 symmetry at a nominal resolution of 2.40 Å, based on the "gold standard" Fourier shell correlation (FSC) = 0.143 criterion (Fig. 1a and Supplementary Fig. 2). The overall atomic model was built based on this density map, which is composed of a PSI core and 16 FCPI subunits surrounding the core (Fig. 1b, c).

The 16 FCPI subunits are named Fcpa1 to Fcpa16 in this study, and are divided into two groups. One group is comprised of Fcpa1–9 and forms a ring around the PSI core and therefore has direct interactions with the core (orange in Fig. 2a), whereas the other group is comprised of Fcpa10–16 attached at the outside of Fcpa6–9 and has no direct interactions with the core (purple in Fig. 2a). Among the 16 FCPI subunits, the positions of Fcpa1, Fcpa5, Fcpa6, and Fcpa7 are similar to those of four LHCR subunits in the red algal PSI-LHCI structures[9,10] (Supplementary Fig. 3a), and the positions of Fcpa1, Fcpa4, Fcpa5, Fcpa6, and Fcpa7 are similar to those of five LHCI subunits in the green algal PSI-LHCI structures[5–8] (Supplementary Fig. 3b). The positions of Fcpa4, Fcpa5, Fcpa6, and Fcpa7 are also similar to those of four LHCI subunits in the plant PSI-LHCI structures[3,4] (Supplementary Fig. 3c). The other Fcpa subunits are unique in the diatom PSI-FCPI structure and do not have corresponding subunits in the plant, green algal, and red algal PSI-LHCI supercomplexes, reflecting the large deviations between the diatom PSI-FCPI and PSI-LHCI supercomplexes from other oxyphototrophs.

**Structure of the PSI core.** The diatom PSI core contains well conserved, seven trans-membrane subunits (PsaA, PsaB, PsaF, PsaI, PsaJ, PsaL, and PsaM) and three stromal subunits (PsaC, PsaD, and PsaE) (Supplementary Fig. 4, 5), and its structure is similar to that of cyanobacterial, algal, and higher plant PSI cores[3–10,16]. However, the diatom PSI core lacked PsaG, PsaH, PsaK, and PsaO subunits, among which, PsaK is present in all PSI cores from cyanobacteria to higher plants, PsaG and PsaH are unique to green lineage organisms, and PsaO is found in eukaryotic oxyphototrophs. The genes of these four PSI subunits are not found in the *C. gracilis* genome as well as the genome of two diatoms *Thalassiosira pseudonana*[17] and *Phaeodactylum tricornutum*[18], reflecting the complete lack of PsaG, PsaH, PsaK, and PsaO in diatom PSI, probably owing to the loss of two genes during the secondary symbiosis (*psaK* and *psaO*) or gaining of two genes in the green lineage (*psaG* and *psaH*). Furthermore, two characteristic subunits were found in the diatom PSI core, which are not present in the structures of plant, green algal, and red algal PSI-LHCI. Among these two subunits, one is a trans-membrane subunit and the other one is an extrinsic subunit located at the stromal side (Fig. 1, and Supplementary Fig. 4, 5). The trans-membrane subunit is attached to PsaB and denoted as Psa28, which has no sequence similarity from other PSI core subunits. Psa28 binds one Chl $a$ molecule at its His110 (Supplementary Fig. 5b). In contrast, the amino-acid residues of the extrinsic subunit could not be assigned, and its structure was modeled as poly-alanines and denoted as Unknown1 (Supplementary Fig. 4, 5c). The structure and binding site of Unknown1 are different from those of ferredoxin as seen in the X-ray crystal structure of ferredoxin-PSI complex from a cyanobacterium *Thermosynechococcus elongatus*[19] (Supplementary Fig. 5d, e). Unknown1 is composed mainly by α-helices, whereas ferredoxin is composed mainly by β-strands surrounded by two short helices from the two sides. Furthermore, Unknown1 tightly associates with PsaD in the PSI-FCPI, whereas ferredoxin is attached to a pocket formed by PsaA, PsaC, and PsaE in the ferredoxin-PSI complex. The whole PSI core in the PSI-FCPI supercomplex contains 94 Chls $a$, 20 β-carotenes, 3 [4Fe-4S] clusters, 2 phylloquinones, and 8 lipid molecules (Supplementary Table 2).

**Structure of FCPI.** The main structures of FCPI subunits are similar to each other and also to those of LHCs. Each FCPI has three major membrane-spanning helices (α1, α2, α3) with several loops and short helices between these three major helices in the

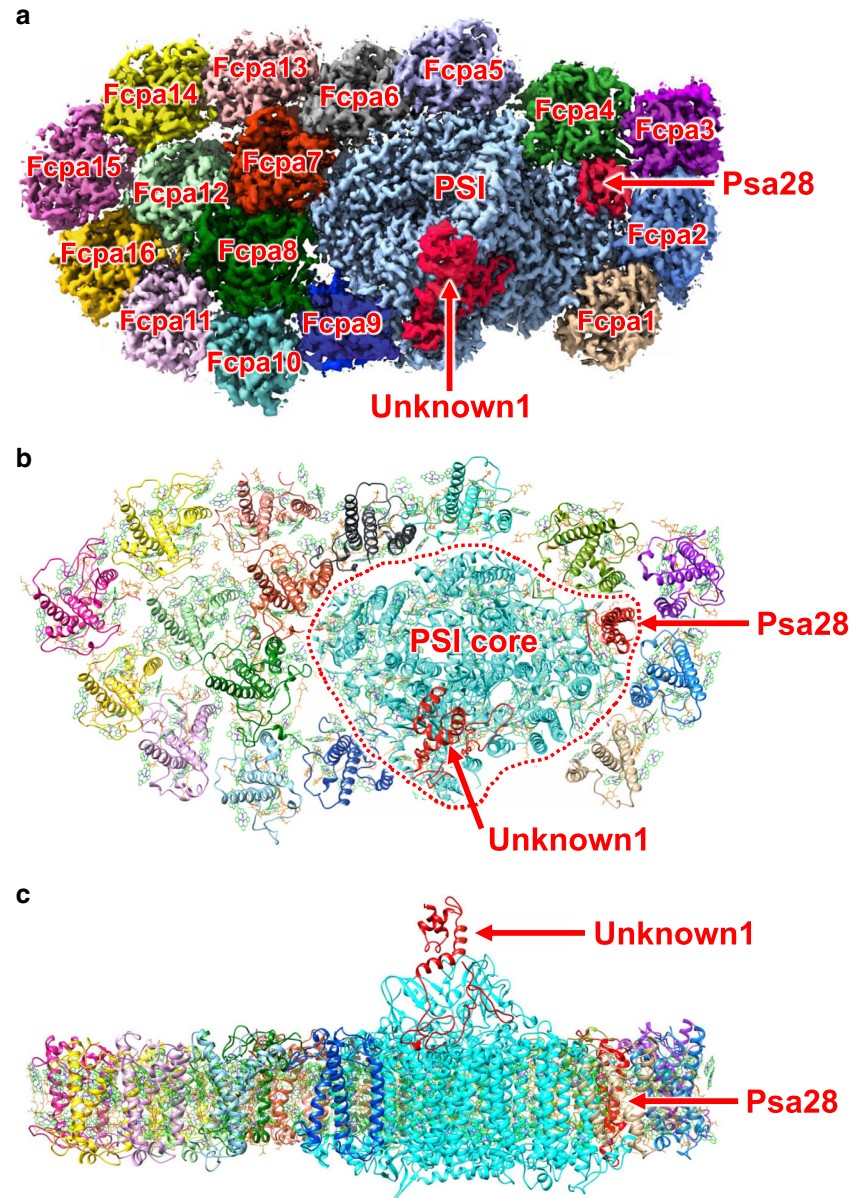

**Fig. 1 Overall structure of the PSI-FCPI supercomplex. a** Cryo-EM density map of the PSI-FCPI with a view from the stromal side. The PSI core is depicted in light blue and labeled as "PSI". Two previously unidentified PSI core subunits are colored red, and labeled as "Unknown1" and "Psa28". The 16 FCPI subunits are labeled as Fcpa1–16. **b** Overall structure of the PSI-FCPI supercomplex with a view from the stromal side (same as in **a**). Red dotted line represents an interface between the PSI core and FCPI. **c** Side view of the PSI-FCPI structure.

stromal and lumenal sides (Fig. 2b and Supplementary Fig. 6, 7). However, a number of differences were found among the structures of the 16 FCPI subunits. In particular, the loop structures of each Fcpa subunit involved in the inter-molecular interactions differ significantly. There are mainly two patterns of interactions between the neighboring Fcpa subunits (Supplementary Fig. 8a). One is a tight association of the N-terminal loop of Fcpa($n$) with the loop between α2 and α3 helices of Fcpa($n+1$) at the stromal side (Supplementary Fig. 8b), and the other one is the interactions between α2 helix of Fcpa($n$) with a loop connecting α1 and α2 helices of Fcpa($n+1$) (Supplementary Fig. 8c). There are also a number of characteristic protein–protein interactions between the inner ring of FCPI and the PSI core (Supplementary Fig. 9a–i). These different interactions may form the basis for the assembly of FCPI, as the different loop structures of the 16 FCPI subunits may determine the binding positions for the individual Fcpa

subunits in the FCPI complex as well as their specific associations with the PSI core.

Draft genome sequences of *C. gracilis* suggested the presence of 43 FCP genes in the *C. gracilis* genome. Phylogenic analysis suggests that these FCP genes can be classified into four types, namely, Lhcr, Lhcf, Lhcx, and Lhcq (Supplementary Fig. 10). The nine FCPs (Fcpa1–9) comprising the inner FCPI ring are classified into the Lhcr family similar to LHCs found in the red algal PSI. Among these FCPs, Fcpa4 (Lhcr4) and Fcpa9 (Lhcr9) tend to be separated into a clade different from other Lhcrs, suggesting their possible different functions. As mentioned above, four Fcpa subunits have binding sites similar to those of LHCRs in the red algal PSI-LHCI[9,10], whereas no binding sites for the remaining five Fcpa subunits exist in the red algal PSI-LHCI[9,10]. Among the seven peripheral FCPIs comprising Fcpa10–16, six (Fcpa11–16) are classified into a unique type of FCP-designated

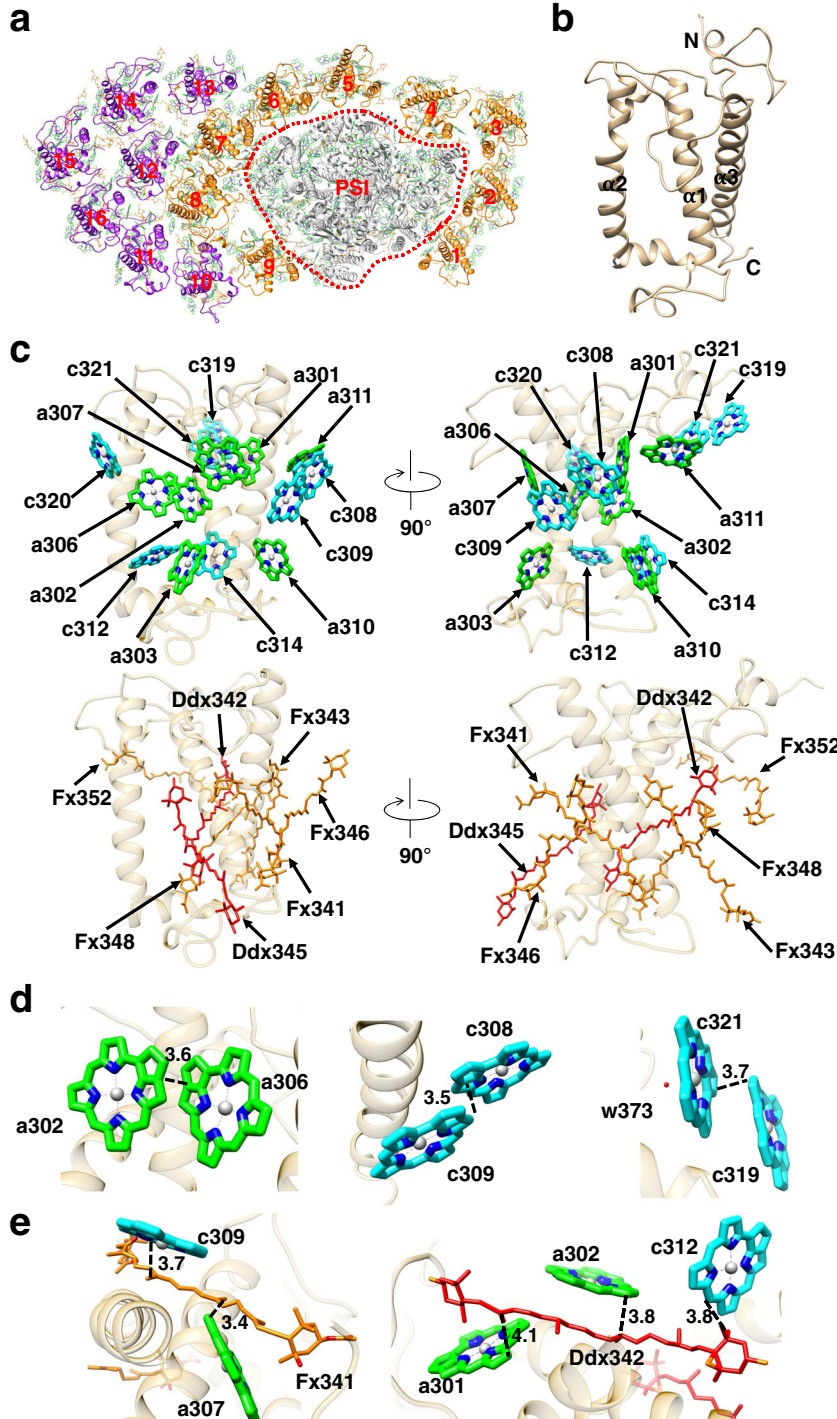

**Fig. 2 Structure of the FCPI. a** Overall structure of the PSI-FCPI with a view from the stromal side. The numbering 1–16 stands for Fcpa1–16 subunits, respectively. Orange-colored FCPI subunits indicate FCPI inner ring surrounding the PSI core, whereas purple-colored FCPI subunits represent peripheral FCPI attached to the outside of the FCPI ring. Red dotted line represents the interface between PSI and FCPI. **b** Structure of the apo-protein of Fcpa8. **c** Arrangement of the pigments (Chls and Cars) within Fcpa8, with the right-side panel rotated 90° clockwise relative to the left-side panel. Green and cyan stand for Chls *a* and *c*, respectively. Orange and red indicate Fx and Ddx, respectively. **d** Three dimeric Chl clusters in Fcpa8. Interactions are indicated by black dashed lines with distances labeled in Å. **e** Characteristic interactions of Cars with Chls within Fcpa8. Interactions are indicated by black dashed lines with distances labeled in Å.

Lhcq, whose functions are not known. The remaining one FCP (Fcpa10) belongs to the Lhcr-type. These peripheral FCPs are not present in red algae and other organisms, implying that during evolution from red algae, diatoms have obtained extra FCP subunits comprising the outer part of the FCPI ring in addition to FCPs that are mainly associated with the PSI core and serve as direct excitation energy donors to the core.

The total number of cofactors bound to the 16 FCPI subunits is 128 Chl *a*, 54 Chl *c*, 54 Fx, 31 Ddx, and 12 lipid molecules (Supplementary Table 3). It should be noted that the structural

differences between the Chl *a* and *c* molecules are the absence of the phytol chain in Chl *c* and an additional difference in the ring structure between the chlorin ring of Chl *a* and the porphyrin ring of Chl *c*, whereas that between Fx and Ddx is the presence of an ester group at one end of Fx (Supplementary Fig. 6). Thus, Chls *a* and *c* can be identified rather easily at the present resolution, whereas identification between Fxs and Ddxs is more difficult given their small differences.

Owing to the differences in the sequences and structures of FCPs, each FCPI subunit has different numbers of pigments (Supplementary Fig. 7 and Supplementary Table 3, 4). For example, Fcpa8 has the largest number of pigments bound (Supplementary Table 3), and its sequence identities and similarities with other FCPI subunits are ranged in 23.8–42.3% and 34.1–54.0%, respectively (Supplementary Table 5). Moreover, the root mean square deviations (RMSD) of Fcpa8 with each of the 15 FCPI subunits are ranged in 1.10–2.47 Å for the $C_\alpha$ atoms (Supplementary Table 5). These differences in the structures and pigments bound among the 16 FCPI subunits may enable each FCP to fulfill their distinct roles in the PSI-FCPI supercomplex.

As Fcpa8 has the largest number of pigments bound among the 16 FCPIs (Supplementary Table 3), we describe the pigment-binding in Fcpa8 in detail. Of the 14 Chl molecules it binds, seven are assigned to Chl *a* (labeled CLA in the PDB file) and seven to Chl *c* (labeled KC1 in the PDB file) (Fig. 2c). These Chls can be clearly categorized into monomeric and dimeric ones. Three pairs of dimeric Chls are found, which are Chls a302/a306, Chls c308/c309, and Chls c319/c321. The edge-to-edge distances within each pair of the dimeric Chls are 3.5–3.7 Å (Fig. 2d). Because the energy level of Chl *a* is lower than that of Chl *c*, the energy level of the dimeric Chl *a* should be lower than that of the dimeric Chl *c*. This indicates that the Chls a302/a306 homo-dimer has the lowest energy level in Fcpa8. Although the Chls a302/a306 and Chls c308/c309 dimers are conserved among the 16 Fcpa subunits, the Chls c319/c321 dimer exists only in Fcpa8 (Supplementary Fig. 11). In addition, a dimeric pair of Chls a303/a313 is formed only in Fcpa7 but not in the other 15 Fcpa subunits (Supplementary Fig. 11). Interestingly, both Chls a303 and a313 are present in Fcpa2 and Fcpa6, but they do not form a dimer due to the different loop structures between Fcpa7 and Fcpa2/6 (Supplementary Fig. 7 and Supplementary Table 4).

Among the seven Cars in Fcpa8, we assigned five Fxs (designated as A86 in the PDB file) and two Ddxs (designated as DD6 in the PDB file) (Fig. 2c, Supplementary Table 3). Among the Fx molecules, Fx341 is closely associated with Chls a307 and c309 with distances of 3.4 and 3.7 Å, respectively (Fig. 2e). Fx343 interacts with Chls c308/a310/c314 at distances ranging in 3.4–5.0 Å. Fx346 is bound to Chl c309 with a distance of 3.5 Å, and Fx348 is associated with Chls a301/a302/a310/c312/c314 at distances of 3.6–5.5 Å. Fx352 interacts with the dimeric Chls c319/c321 at distances of 3.7–5.0 Å. Ddx342 is tightly coupled to Chls a301/a302/c312 at distances of 3.8–4.1 Å (Fig. 2e), whereas Ddx345 interacts with Chls a303 and a306 at distances of 3.4–6.0 Å. These results suggest close interactions of Fxs and Ddxs with Chls *a* and *c*.

**Interactions of Chls among FCPI subunits**. The 16 FCPI subunits also interact with each other through extensive pigment–pigment interactions, which are important for EET among different FCPs. Based on the Chl-Chl distances, a number of possible EET pathways can be proposed within the peripheral FCPIs (Fig. 3), from the peripheral FCPIs to the inner FCPI ring (Fig. 3), and within the inner FCPI ring (Fig. 4). In the peripheral FCPI, EETs between Fcpa16 and Fcpa11, Fcpa12, Fcpa15 are possible through interactions between Fcpa16-c328 and Fcpa11-

c323 (4.9 Å), between Fcpa16-a322 and Fcpa12-a308 (9.0 Å), and between Fcpa16-a322 and Fcpa15-a302 (11.8 Å) (Fig. 3b, c). EETs between Fcpa15 and Fcpa14, Fcpa12 may occur through interactions between Fcpa15-a330 and Fcpa14-a308 (5.1 Å), and between Fcpa15-a328 and Fcpa12-a308/c309 (13.9–14.0 Å) (Fig. 3c). Fcpa14 is associated with Fcpa13 and Fcpa12 at 5.1 Å (Fcpa14-a330 and Fcpa13-a308) and 12.8 Å (Fcpa14-a329 and Fcpa12-c304) (Fig. 3d). Fcpa13 is associated with Fcpa12 at 9.0 Å (Fcpa13-a316 and Fcpa12-c304) (Fig. 3d). Fcpa12 is associated with Fcpa11 at 10.3 Å (Fcpa12-c316 and Fcpa11-c304) (Fig. 3b), whereas Fcpa11 is associated with Fcpa10 at 15.3–15.6 Å (Fcpa11-c302 and Fcpa10-c309/a303) (Fig. 3e).

In the interface between the peripheral FCPI and the inner FCPI ring, Fcpa13 is tightly coupled with Fcpa6 and Fcpa7 at distances of 4.6–5.8 Å (Fcpa13-c325/c326 and Fcpa6-a308/c309) and 14.0–14.2 Å (Fcpa13-a302/a303 and Fcpa7-a303/a309/a313) (Fig. 3f). Fcpa12 is closely associated with Fcpa7 and Fcpa8 at distances of 4.4–5.6 Å (Fcpa12-a306/a315/c325 and Fcpa7-c308/c318) and 14.0–14.5 Å (Fcpa12-a302/c325 and Fcpa8-a303/a307/c309) (Fig. 3g). Fcpa11-a306 interacts with Fcpa8-c308 at 7.4 Å (Fig. 3e). Fcpa10 interacts with Fcpa8 at 4.3–8.8 Å (Fcpa10-a307/c304 and Fcpa8-a310/a311), and with Fcpa9 at 9.4 Å (Fcpa10-c312 and Fcpa9-c310) (Fig. 3h).

In the inner FCPI ring (Fig. 4), Chl a308 in Fcpa9 is tightly coupled with Chl c319 in Fcpa8 (4.9 Å) (Fig. 4b). Fcpa8 is associated with Fcpa7 through interactions between Fcpa8-a307 and Fcpa7-a311, and between Fcpa8-a303 and Fcpa7-c318 at distances of 8.0–8.4 Å (Fig. 4c). Fcpa7 interacts with Fcpa6 at 7.9–8.5 Å (between Fcpa7-a307 and Fcpa6-c311, and between Fcpa7-a313 and Fcpa6-c310) (Fig. 4d). Fcpa6 is closely associated with Fcpa5 at 3.6–7.8 Å (Fcpa6-a304/a307/a315 and Fcpa5-a310/c305/c311) (Fig. 4e). Chl c304 in Fcpa5 interacts with Chl c316 in Fcpa4 at 4.2 Å (Fig. 4f), and Chl a317 in Fcpa4 is associated with Chl a302 in Fcpa3 at 10.8 Å (Fig. 4g). Fcpa3 is associated with Fcpa2 at 5.2–9.5 Å (between Fcpa3-a307 and Fcpa2-c311, and between Fcpa3-c304 and Fcpa2-a310) (Fig. 4h). Fcpa2 interacts with Fcpa1 through interactions between Fcpa2-a307 and Fcpa1-c311, and between Fcpa2-c304 and Fcpa1-a310, at 5.7–8.4 Å (Fig. 4i).

**Chl interactions between FCPIs and PSI core**. Some of the FCPI subunits in the inner FCPI ring are associated with the PSI cores directly through pigment–pigment and pigment–protein interactions, enabling EET possible from FCPIs to the PSI core efficiently (Fig. 5). Fcpa9 interacts with PsaA and PsaL at distances ranging in 4.5–8.8 Å (between Fcpa9-a307/c322 and PsaA-a837/a846, and between Fcpa9-a304 and PsaL-a203) (Fig. 5b). Fcpa8 is associated with PsaA at 5.2–8.9 Å (Fcpa8-c312/c320 and PsaA-a821/a823/a845) (Fig. 5c). Fcpa7-a302 interacts with PsaA-a813/a821/a845 at 12.2–12.7 Å (Fig. 5d). Fcpa6 is associated with PsaA and PsaJ at 7.0–11.5 Å (between Fcpa6-a312/a314 and PsaA-a808/a817, and between Fcpa6-a315 and PsaJ-a101) (Fig. 5e). Fcpa5 is associated with PsaF and PsaJ at 7.1–12.8 Å (between Fcpa5-a302 and PsaF-a201, and between Fcpa5-c305 and PsaJ-a101) (Fig. 5f). Fcpa4-a306 is tightly coupled with Chl a201 of Psa28 at 5.2 Å and also with PsaB-a842 at 9.0 Å (Fig. 5g). Fcpa2-a306 is associated with PsaB-a820 in PsaB at 6.9 Å (Fig. 5h), and Fcpa1-a302 and a306 interact with PsaB-a811 at 6.4 and 8.6 Å, respectively (Fig. 5i).

**Possible EET pathways in PSI-FCPI**. Based on the pigment–pigment interactions described above (Figs. 3, 4, 5), possible EET pathways in the diatom PSI-FCPI are summarized in Supplementary Fig. 12. Within the peripheral FCPI (Fcpa10–16), inter-subunit EET pathways include Fcpa16→Fcpa11/12/15,

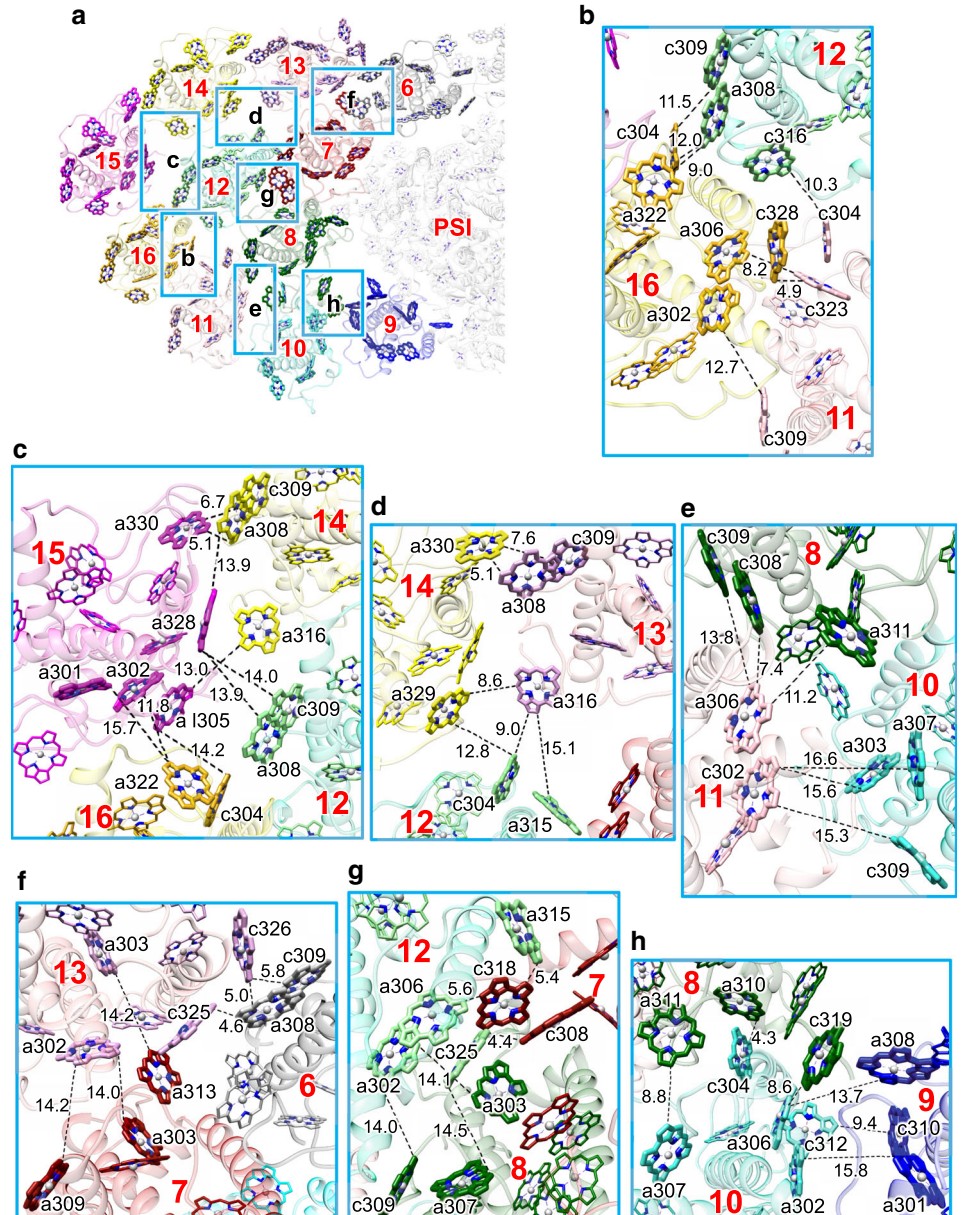

**Fig. 3 Chl-Chl interactions within the peripheral FCPI and between the peripheral and inner ring FCPIs. a** Interfaces among different FCP units with a view from the stromal side. Squared areas are enlarged in **b**–**h**. The numbering 6–16 stands for Fcpa6–16 subunits, respectively. **b** Interactions between Fcpa11/12 and Fcpa16. **c** Interactions between Fcpa12/14/15 and Fcpa16. **d** Interactions between Fcpa12/13 and Fcpa14. **e** Interactions between Fcpa8/10 and Fcpa11. **f** Interactions between Fcpa6/7 and Fcpa13. **g** Interactions between Fcpa7/8 and Fcpa12. **h** Interactions between Fcpa8/9 and Fcpa10. For clarity, only Chl-Chl interactions are shown, whereas the protein–protein and protein–pigment interactions are omitted. Interactions are indicated by dashed lines with distances labeled in Å.

Fcpa15→Fcpa12/14/16, Fcpa14→Fcpa12/13/15, Fcpa13→Fcpa12/14, Fcpa12→Fcpa11/13/14/15/16, Fcpa11→Fcpa10/12/16 and Fcpa10→Fcpa11. From the peripheral FCPI to Fcpa6–9 in the inner FCPI ring, possible EET pathways include Fcpa13→Fcpa6/7, Fcpa12→Fcpa7/8, Fcpa11→Fcpa8 and Fcpa10→Fcpa8/9. Within the inner FCPI ring, EET is possible between each pair of the adjacent FCPI subunits, e.g., between Fcpa(*n*) and Fcpa(*n* + 1) (There EET pathways are: Fcpa1 ↔ Fcpa2, Fcpa2 ↔ Fcpa3, Fcpa3 ↔ Fcpa4, Fcpa4 ↔ Fcpa5, Fcpa5 ↔ Fcpa6, Fcpa6 ↔ Fcpa7, Fcpa7 ↔ Fcpa8 and Fcpa8 ↔ Fcpa9). The time constant of the EET of the peripheral FCPI to the FCPI ring, and the energy migrations within the FCPI ring, may be at ~600 fs, because the EET from a pigment pool of Chls *a* and *c* to other Chl molecules has been found at time constants of 550–660 fs[12].

The energy absorbed by FCPI will finally be transferred and trapped at the PSI reaction center. From the inner FCPI ring to the PSI core, possible EET pathways are found as Fcpa9→PsaA/L, Fcpa8→PsaA, Fcpa7→PsaA, Fcpa6→PsaA/J, Fcpa5→PsaF/J, Fcpa4→PsaB/28, Fcpa2→PsaB, and Fcpa1→PsaB. The energy of FCPI is likely transferred to the PSI core within 2 ps by our recent femtosecond time-resolved fluorescence (TRF) spectro-scopic analysis[12], which occurs after the energy migration among the FCPI within 600 fs.

**Energy quenching sites in PSI-FCPI.** Cars play dual roles in the EET events: one is the energy transfer to Chl molecules, and the other one is the energy quenching. The latter event is often facilitated by Chl–Car interactions in LHCs and photosystem

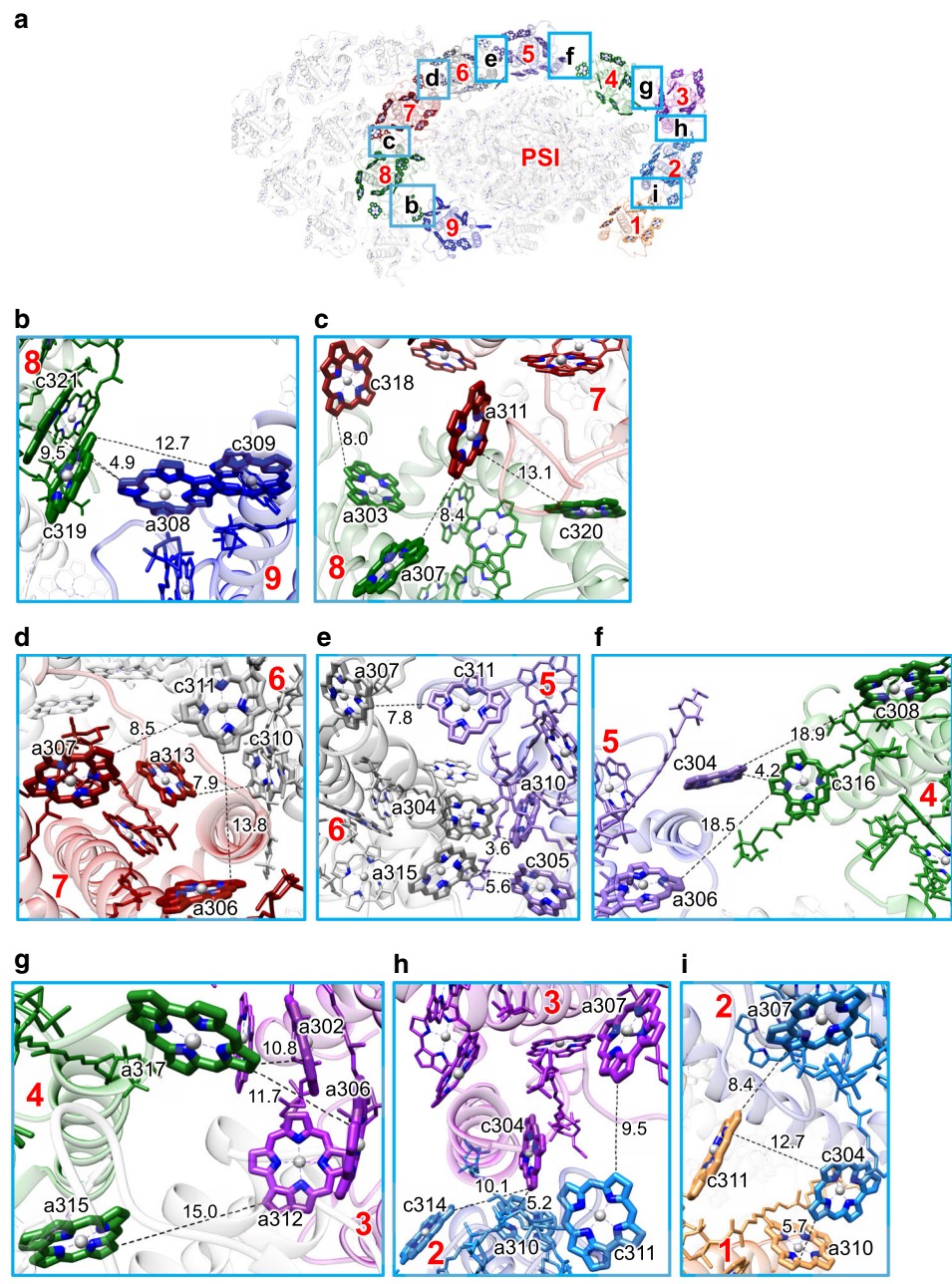

**Fig. 4 Chl-Chl interactions within the inner FCPI ring. a** Interfaces among different FCP units with a view from the stromal side. Squared areas are enlarged in **b**–**i**. The numbering 1–9 stands for Fcpa1–9 subunits, respectively. **b** Interactions between Fcpa8 and Fcpa9. **c** Interactions between Fcpa7 and Fcpa8. **d** Interactions between Fcpa6 and Fcpa7. **e** Interactions between Fcpa5 and Fcpa6. **f** Interactions between Fcpa4 and Fcpa5. **g** Interactions between Fcpa3 and Fcpa4. **h** Interactions between Fcpa2 and Fcpa3. **i** Interactions between Fcpa1 and Fcpa2. For clarity, only Chl-Chl interactions are shown, whereas the protein–protein and protein–pigment interactions are omitted. Interactions are indicated by dashed lines with distances labeled in Å.

cores[20,21]; therefore, possible quenching sites can be examined based on the distances and orientations of Chl–Car interactions. A number of Chl–Car interactions within/around FCPI are found in the present PSI-FCPI structure (Figs. 2, 6). As described above, each FCPI subunit has several close interactions between Chls *a*/*c* and Fx/Ddx, albeit with different numbers of pigments bound among the different FCPI subunits (Fig. 2, Supplementary Fig. 7, and Supplementary Table 3, 4).

In addition to Chl–Car interactions within each FCPI subunit, the interfaces between adjacent FCPI subunits possess characteristic Chl–Car interactions (Fig. 6). Fcpa16 is closely associated with Fcpa11 and Fcpa15 through interactions between Fcpa16-

a306/c328 and Fcpa11-Fx355, and between Fcpa16-a322 and Fcpa15-Ddx344, at 3.4–8.4 Å (Fig. 6b). Fx358 of Fcpa15 interacts with Chl a316 of Fcpa14 (8.8 Å) (Fig. 6c). Fcpa14 interacts with Fcpa12 between Fcpa14-a329/Fx344 and Fcpa12-a324/Ddx345 at 6.7–8.4 Å (Fig. 6c). Fcpa13 interacts with Fcpa7 and Fcpa12 between Fcpa13-a302 and Fcpa7-Fx351, and between Fcpa13-Fx343 and Fcpa12-c304, at 4.7–4.9 Å (Fig. 6d). Fcpa12 interacts with Fcpa8 and Fcpa11 between Fcpa12-a302 and Fcpa8-Fx346/Ddx345, and between Fcpa12-Fx343 and Fcpa11-c304, at 5.2–7.1 Å (Fig. 6e). Chl c304 of Fcpa10 interacts with Fx343 of Fcpa8 at 4.2 Å (Fig. 6f). Chl a304 of Fcpa6 is tightly coupled with Ddx343 of Fcpa5 at 3.6 Å (Fig. 6g). Chl c304 of Fcpa5 interacts

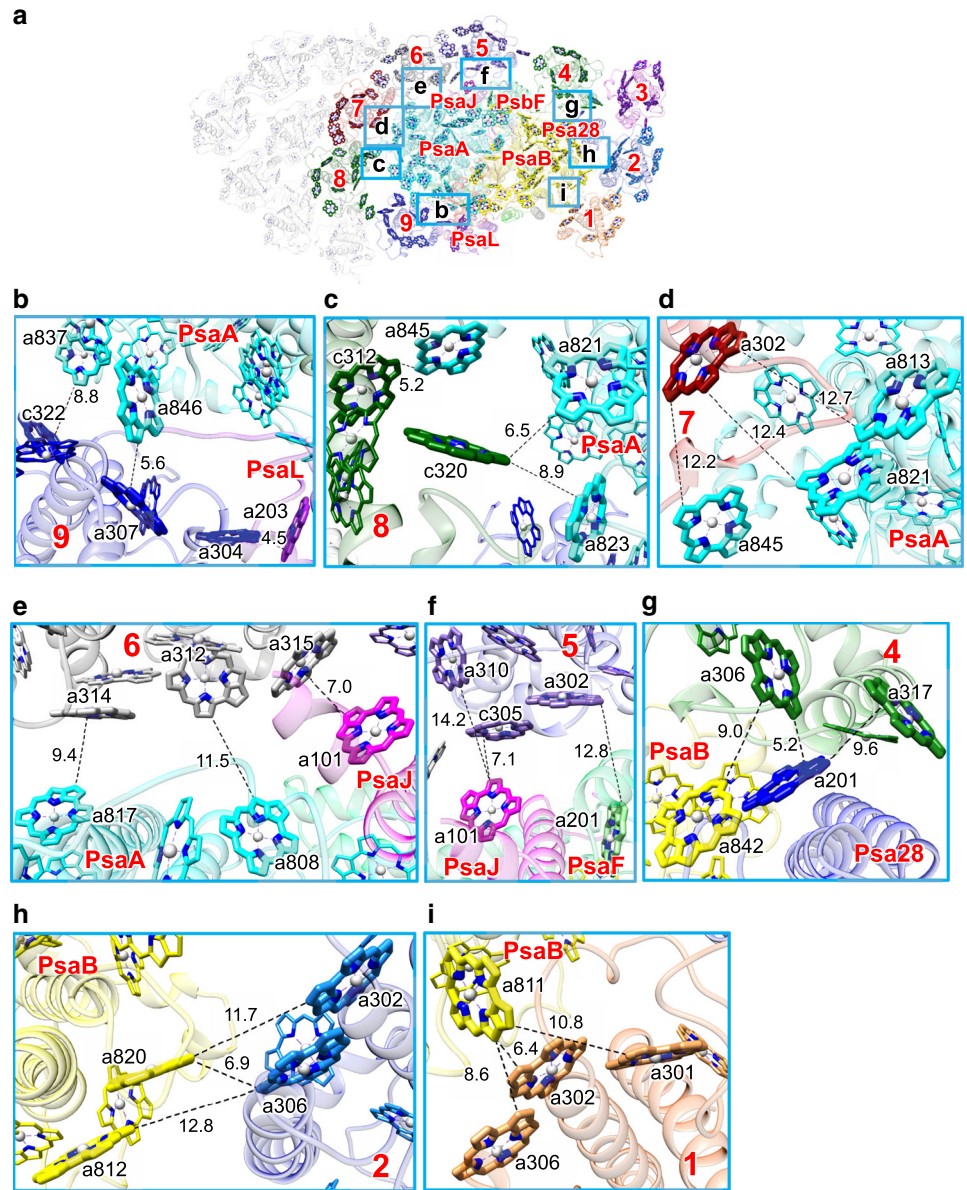

**Fig. 5 Chl-Chl interactions between the ring FCPIs and PSI core. a** Interfaces between Fcpa and PSI subunits with a view from the stromal side. Squared areas are enlarged in **b**–**i**. The numbering 1–9 stands for Fcpa1–9 subunits, respectively. **b** Interactions between Fcpa9 and PsaA/PsaL. **c** Interactions between Fcpa8 and PsaA. **d** Interactions between Fcpa7 and PsaA. **e** Interactions between Fcpa6 and PsaA/PsaJ. **f** Interactions between Fcpa5 and PsaF/PsaJ. **g** Interactions between Fcpa4 and PsaB/Psa28. **h** Interactions between Fcpa2 and PsaB. **i** Interactions between Fcpa1 and PsaB. For clarity, only Chl-Chl interactions are shown, whereas the protein–protein and protein–pigment interactions are omitted. Interactions are indicated by dashed lines with distances labeled in Å.

with Fx343 of Fcpa4 at 7.2 Å (Fig. 6h). Chl c304 of Fcpa3 is tightly coupled with Ddx343 of Fcpa2 at 3.9 Å (Fig. 6i). Chl c304 of Fcpa2 closely interacts with Fx343 of Fcpa1 at 3.5 Å (Fig. 6j).

Chl–Car interactions are also found in the interfaces between FCPI and PSI. Fcpa9-a304 interacts with PsaL-BCR212 at 4.9 Å, and Fcpa9-Fx341 interacts with PsaA-a846 at 4.2 Å (Fig. 6k). Fcpa8-c320 interacts with PsaA-BCR861 at 5.0 Å (Fig. 6l). Fcpa7 interacts with PsaA between Fcpa7-Fx343 and PsaA-a845, and between Fcpa7-Ddx350 and PsaA-a811, at 5.9–8.6 Å (Fig. 6l). Fcpa6-Ddx343 is associated with PsaA-a817 at 5.1 Å (Fig. 6m), and Fcpa4-Fx349 interacts with PsaB-a842 at 4.3 Å (Fig. 6n).

The close interactions between Chls-Fxs appear to be a common feature of FCPs[22–24]. Based on the well-known quenching mechanisms by Chl–Car interactions[20,21], it can be proposed that the energy quenchers are Fx and Ddx in FCPI. Ddx

is converted to diatoxanthin (Dtx) under high-light conditions through a Ddx cycle, which may function as a quencher for excitation energy[25]. However, we have shown that both Ddx and Fx are plausible candidates for the energy quencher as studied by TRF spectroscopy using isolated FCP complexes, PSII-FCPII and PSI-FCPI supercomplexes purified from the diatom grown at a photosynthetic photon flux density of 30 μmol photons m$^{-2}$ s$^{-1}$, because Dtx was not observed in these complexes[13,14,26,27]. This indicates that Dtx may not be the energy quencher under low-light conditions, under which oxygen-evolving activity of PSII is not impaired[28]. The Dtx-independent quenching observed in PSI-FCPI may be similar to the structural and spectroscopic observations of the PSII-FCPII supercomplexes[23,27], indicating the functioning of this quenching mechanism in the PSI-FCPI and PSII-FCPII of diatoms under low-light conditions.

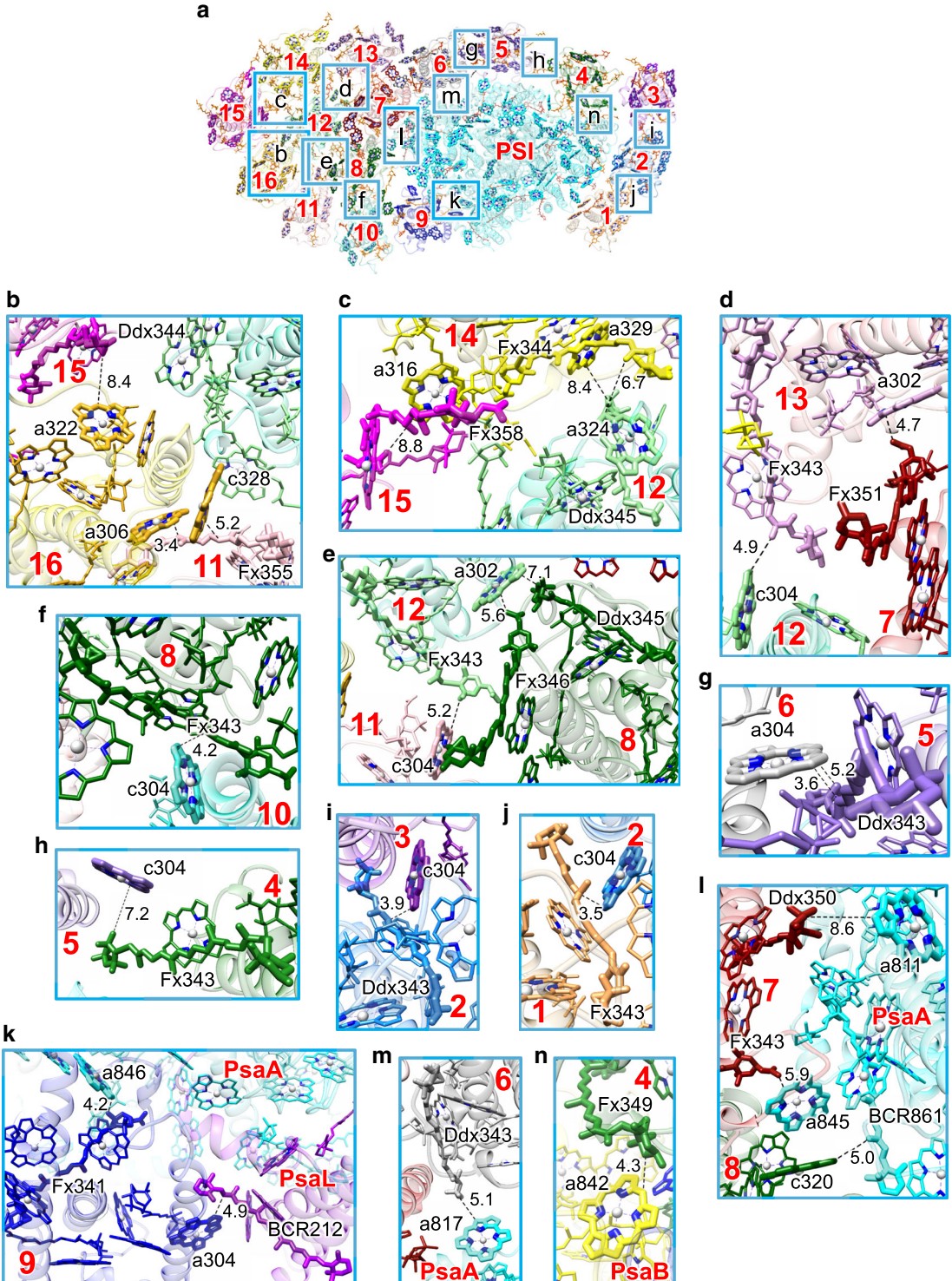

**Fig. 6 Chl–Car interactions in the PSI-FCPI supercomplex. a** Interfaces in the PSI-FCPI with a view from the stromal side. Squared areas are enlarged in **b–n**. The numbering 1–16 stands for Fcpa1–16 subunits, respectively. **b** Interactions between Fcpa11/15 and Fcpa16. **c** Interactions between Fcpa12/14 and Fcpa15. **d** Interactions between Fcpa7/12 and Fcpa13. **e** Interactions between Fcpa8/11 and Fcpa12. **f** Interactions between Fcpa8 and Fcpa10. **g** Interactions between Fcpa5 and Fcpa6. **h** Interactions between Fcpa4 and Fcpa5. **i** Interactions between Fcpa2 and Fcpa3. **j** Interactions between Fcpa1 and Fcpa2. **k** Interactions between Fcpa9 and PsaA/PsaL. **l** Interactions between Fcpa7/8 and PsaA. **m** Interactions between Fcpa6 and PsaA. **n** Interactions between Fcpa4 and PsaB. For clarity, only Chl–Car interactions are shown, whereas the protein–protein and protein–pigment interactions are omitted. Interactions are indicated by dashed lines with distances labeled in Å.

It is widely accepted that excess light leads to acidification of the thylakoid lumen[29], which results in enhancement of energy quenching in the green lineages[30,31]. Energy quenching in diatom PSI-FCPI is also facilitated under the acidic pH conditions as revealed by TRF spectroscopy[14], suggesting that the structural changes by protonation of either amino-acid residues such as Asp and Glu or pigment itself result in the enhancement of the quenching capacity. In both lumenal and stromal sides in the PSI-FCPI structure, there are a number of interactions between Asp/Glu and pigments. At the lumenal side, some Asp and Glu residues are located near the Chl–Car pairs of Chl $a$/Ddx and Chl $c$/Fx, which are observed in Fcpa2, Fcpa3, Fcpa6, Fcpa9, Fcpa10, Fcpa14, and Fcpa15 (Supplementary Fig. 13). These Asp/Glu-Chls/Cars interactions may be related to the quenching events in response to the pH changes induced by light illumination.

## Discussion

The structure of PSI-LHCI supercomplexes has been solved from various oxyphototrophs (Supplementary Fig. 3)[3–10]. In the green lineage, four LHCI subunits are found in the plant PSI-LHCI[3,4], whereas four-ten LHCI subunits are found in the green algal PSI-LHCI[5–8]. By contrast, three or five LHCR subunits are found in the red algal PSI-LHCI[9,10]. The diatom PSI-FCPI binds 16 FCPI subunits, which are the largest number of LHCI associated with PSI reported so far. These FCPI subunits are organized into an inner, completely closed ring surrounding the PSI core, and two semi-layers at the outside of the inner ring. It is particularly interesting to note that all of the FCPI inner ring components, Fcpa1–9 belong to the Lhcr family found in the red algae, whereas the peripheral FCPI components, Fcpa11–16, belong to a unique class of FCP, the Lhcq family (except Fcpa10, which belongs to Lhcr). The number of Lhcr subunits found in the diatom PSI-FCPI is double of that in the red algal PSI-LHCI, indicating the acquisition of the additional LHCR subunits in the diatom PSI-FCPI during its evolution from red algae. Furthermore, diatoms have acquired the distinctive Lhcq protein family during evolution. The positions of Fcpa1/5/6/7 (Lhcrs) in the diatom PSI-FCPI correspond to those of Lhcr2*/1/2/3 in the red algal PSI-LHCI, respectively (Supplementary Fig. 3), indicating their evolutionary relationship. By contrast, the green algal LHCI subunits, Lhca9/1a/8/7/3, are located in positions similar to those of the Fcpa1/4/5/6/7, respectively, whereas the positions of the plant LHCI subunits, Lhca3/2/4/1, are similar to those of the Fcpa4/5/6/7, respectively (Supplementary Fig. 3). However, the green type LHCI subunits cannot be superposed with the diatom FCPI subunits exactly (Supplementary Fig. 3). In addition, there are no Fcpa subunits at positions corresponding to Lhcr1* in the red algal PSI-LHCI and Lhca2 in the green algal PSI-LHCI. These structural observations imply that the binding sites and properties of LHCIs differ significantly between the red and green lineage organisms.

It is known that in the green lineages, the structures of LHCI are highly similar to those of LHCII. The structures of FCPI are compared with diatom FCPII, red algal LHCR, plant LHCII, and LHCI (Supplementary Fig. 14–17). The RMSD of the FCPI with the FCPII, LHCR, LHCII, and LHCI are 1.83–2.21 Å. Most of the Chl and Car-binding sites in LHCR are conserved in the FCPI, whereas these sites in FCPII, LHCII, and LHCI are similar to those in FCPI to a lesser extent. These observations suggest that the protein structure and pigment-binding sites are well conserved in the LHCI family in the red lineages. However, it is interesting to note that the dimeric Chl sites of Chls a302/a306 and Chls a308/c309 in FCPI are completely conserved in the FCPII, LHCR, LHCII, and LHCI, implying that oxyphototrophs may rely on the dimeric Chls for light-harvesting strategy

including EET and quenching, irrespective of the red and green lineages.

Alterations in some of the PSI core components are also found in the diatom PSI-FCPI (Supplementary Fig. 3), which may be responsible for the unique and efficient associations of the 16 FCPI subunits with the PSI core. The red algal PSI-LHCI has PsaK and PsaO, which are not found in the diatom PSI-FCPI, implying that the lack of PsaK and PsaO enables the binding of both Fcpa8 and Fcpa9 to the PSI core in the diatom PSI-FCPI. Moreover, the diatom PSI-FCPI structure has no PsaG and PsaH as seen in the plant and green algal PSI-LHCI (Supplementary Fig. 3). Instead, a previously unidentified subunit Psa28 was found in the diatom PSI-FCPI, which is located at a position similar to that of PsaG; this subunit appears to support the association of Fcpa2–4 with the PSI core. Thus, the differences of PSI core subunits among oxyphototrophs may lay the foundation for the assembly of LHCs in the PSI-LHCI supercomplexes.

The increased number of LHC subunits in diatoms confer them with special capabilities of light-harvesting and energy quenching under the aquatic environment conditions, where light is often limited and its intensity is highly fluctuated. It is also interesting to note that the fluorescence properties of *C. gracilis* cells are strongly changed under different growth conditions, such as $CO_2$ concentration, temperature, and light intensity[32–36], suggesting that the expression of FCPs strongly depends on the growth conditions. Actually, the molecular sizes of the PSI-FCPI supercomplexes were changed under these conditions[34], implying that the FCPI subunits are markedly regulated in response to the growth conditions. These observations imply that the expression of FCPs in diatoms is highly dynamic and regulated in order to adapt to the specific growth environments of the organisms. These special features may have greatly contributed the success of diatoms in the aquatic environments.

## Methods

**Purification of the PSI-FCPI supercomplex from *C. gracilis*.** The marine centric diatom, *C. gracilis* (UTEX LB 2658) was grown in artificial seawater[37] at a photosynthetic photon flux density of 30 µmol photons $m^{-2} s^{-1}$ at 30℃ with continuous bubbling of air containing 3% (v/v) $CO_2$[11,23,34]. The PSI-FCPI supercomplexes were purified using sucrose density gradient centrifugation[11] and concentrated using a 100 kDa cutoff filter (Amicon Ultra; Millipore) at 4000 × $g$ with a buffer containing 40 mM sucrose, 20 mM 2-(N-morpholino)ethanesulfonic acid (MES)-NaOH (pH 6.5) and 0.02% (w/v) $n$-dodecyl-$\beta$-D-maltoside ($\beta$-DDM). Polypeptide profile, spectroscopic analyses and pigment composition of the supercomplex have been reported previously[11,13].

**Polypeptide assignment of PSI-FCPI by mass spectrometry.** The CBB-stained gel bands were cut out and alkylated with iodoacetate. The gel slices were treated with either trypsin (TPCK treated; Worthington Biochemical) or chymotrypsin (Chymotrypsin, Alpha, Purified; Worthington Biochemical), and the resultant peptide samples were subjected to LC-MSMS analysis using Easy nLC 1000 and Q Exactive (Thermo Fisher Scientific). The mass spectra obtained were analyzed[23].

**Phylogenic and sequence analyses.** Polypeptide sequences of the FCPs were collected from the draft genome data of *C. gracilis*[38]. Multiple sequence alignment was constructed using MAFFT[39]. After manual refinement of the alignment, maximum-likelihood (ML) trees were constructed using RAxML with 1000 bootstrap resamplings[40], and the phylogenic tree obtained was visualized using FigTree v1.4.4 (https://github.com/rambaut/figtree/releases).

**Cryo-EM data collection.** For cryo-EM observations, 1-µL aliquots of the PSI-FCPI supercomplexes (2 or 4 mg of Chl $mL^{-1}$) in a buffer containing 20 mM MES-NaOH (pH 6.5), 40 mM sucrose and 0.02% β-DDM were applied to holey carbon grids (Quantifoil R1.2/1.3 Mo 300 mesh or Quantifoil R2/1 Mo 300 mesh) at 4℃ with 100% humidity. The grids were plunged into liquid ethane cooled by liquid nitrogen after blotting with filter papers for 15 s using a Vitrobot mark IV (Thermo Fisher Scientific), and then transferred into a cryo-electron microscope (Titan Krios; Thermo Fischer Scientific) operated at 300 kV. The cryo-electron microscope was equipped with a field emission gun, a Cs corrector (CEOS GmbH) and a direct electron detection camera (Falcon 3EC; Thermo Fischer Scientific). Movies were recorded at a nominal magnification of ×59,000 using the Falcon 3EC in

linear mode (a calibrated pixel size of 1.113 Å). The nominal defocus range was −1.5 to −3.0 μm. Each exposure of 50 electrons Å$^{-2}$ for 2.5 s was dose-fractionated into 33 movie frames.

**Cryo-EM image processing.** The movie frames were aligned and summed using the MotionCor2 software[41] to obtain a final dose-weighted image. Estimation of the contrast transfer function (CTF) was performed using the CTFFIND4 program[42]. All micrographs exhibited good power spectra over 5 Å spacial resolution based on the extent of the Thon rings. All of the following processes were performed using RELION-3[43]. In total 2,689,965 particles were automatically picked from 9,910 micrographs and then used for reference-free 2D classification. The resultant 1,654,299 particles from the good 2D classes were subjected to 3D classification with a C1 symmetry. All 3D classification processes described below was performed with a C1 symmetry. For making an initial 3D model of the PSI-FCPI, a part of the particles from good 2D classes was applied to 3D refinement with an initial model of a cryo-EM structure of a green algal PSI-LHCI complex (EMD-9853 [https://www.emdataresource.org/EMD-9853]) with a 60 Å low-pass filter. After post-processing, an 8.6 Å resolution map of the PSI-FCPI was obtained, which was employed for the initial model used for the first 3D classification of all particles in the good 2D classes. As described in Supplementary Fig. 1c, 790,522 particles (class II) were selected and subjected to subsequent 3D refinement and post-processing, and then to CTF refinement and Bayesian polishing. After applying the polished particles to 3D refinement and post-processing, the 790,522 particles were subjected to a second-round 3D classification with a mask of the peripheral FCPI region, which resulted in 470,801 particles (class 5) based on a good map quality with ~60% of the total particles among six classes. These good particles were subjected again to 3D refinement and post-processing, and then to CTF refinement and Bayesian polishing. After applying the polished particles to 3D refinement and post-processing, the PSI-FCPI structure was reconstructed at 2.40 Å resolution based on the gold-standard FSC technique with a cutoff value of 0.143[44]. In the resultant overall map of the PSI-FCPI, the peripheral region of FCPI, especially Fcpa15, has weak densities; therefore, the PSI core map was subtracted from the overall map in order to refine the peripheral FCPI region. The subtracted particles were subjected to 3D refinement and post-processing, and the peripheral FCPI map was reconstructed at 2.60 Å resolution. The local resolution of the final maps was calculated using RELION.

**Model building and refinement.** The 2.6-Å peripheral FCPI map was used for model building of Fcpa6, Fcpa7, Fcpa8, Fcpa10, Fcpa11, Fcpa12, Fcpa13, Fcpa14, Fcpa15, and Fcpa16. For the model building of Fcpa subunits, individual chains were first traced in Cα baton mode with Coot[45]. All Fcpa subunits were assigned and built on the basis of interpretable features from the density map, including regions enriched in bulky residues and axial ligands of Chls. For the assignment of Chls, Chls $a$ and $c$ were distinguished by inspection of the density map corresponding to the phytol chain with a threshold of 12 σ, which was found to be the least level not to link the map of Chls with that of noise. All Chls $c$ were assigned as Chl $c1$, because of difficulty in distinction between Chl $c1$ and Chl $c2$ at the present resolution. For the assignment of carotenoids, Fx and Ddx were distinguished on the basis of the density covering the head group of carotenoids with a threshold of 10 σ as described above. The peripheral FCPI structure was then refined with phenix.real_space_refine[46] with geometric restraints for the protein–cofactor coordination. The final model was further validated with MolProbity[47] and EMringer[48].

The 2.4-Å PSI-FCPI map was used for model building of the overall PSI-FCPI supercomplex. For model building of the PSI core, each subunit of the homology models constructed using the Phyre2 server[49] was first manually fitted into the 2.4-Å cryo-EM map in UCSF Chimera[50], and then adjusted in Coot. For the model building of the other Fcpa subunits (Fcpa1, Fcpa2, Fcpa3, Fcpa4, Fcpa5, and Fcpa9), the models of individual chains were built in a similar manner to the peripheral FCPI. For the assignment of Chls and carotenoids, Chls $a/c$ and Fx/Ddx were distinguished in a similar procedure as the peripheral FCPI with a threshold of 9 σ and 7 σ, respectively. Finally, the model for the PSI-FCPI supercomplex was assembled by fitting the peripheral FCPI model to the 2.4-Å map with UCSF Chimera. The complete PSI-FCPI supercomplex structure was then refined in a similar manner as the peripheral FCPI. The final model was further validated with MolProbity and EMringer. The statistics for all data collection and structure refinement are summarized in Supplementary Table 1.

**Statistics and reproducibility.** As described in the "Methods" section, numerous PSI-FCPI particles were picked up from the cryo-EM images and used for the structural analysis with standard protocols, and the data statistics, evaluation of the resolution are documented in Supplementary Fig. 1, 2 and Supplementary Table 1.

**Reporting summary.** Further information on research design is available in the Nature Research Reporting Summary linked to this article.

## Data availability

Atomic coordinates and cryo-EM maps for the reported structure of PSI-FCPI and peripheral FCPI have been deposited in the Protein Data Bank under accession codes 6L4U and 6L4T, respectively, and in the Electron Microscopy Data Bank under accession codes EMD-0835 and EMD-0834, respectively. Other data are available from the corresponding authors upon reasonable request.

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

## Acknowledgements

This work was supported by the Platform Project for Supporting Drug Discovery and Life Science Research (Basis for Supporting Innovative Drug Discovery and Life Science Research (BINDS)) from AMED, PRESTO from JST grant No. JPMJPR16P1 (F.A.), JSPS KAKENHI grant Nos. JP17K07442, JP19H04726 (R.N.), JP16H06553 (S.A.), JP17H06433 (J.-R.S.), Advanced Low Carbon Technology Research, and Development Program from the Japan Science and Technology Agency grant No. JPMJAL1105 (Y.K. and K.I.), the joint usage/research program of the Artificial Photosynthesis Osaka City University (R.N.), and a Collaborative Research Program from National Institute for Basic Biology grant No. 19-455 (K.I.). We thank Ms. Hiroyo Nishide, NIBB, for supporting the genome data analysis.

## Author contributions

R.N., N.M., J.-R.S., and F.A. conceived the project; R.N. purified the PSI-FCPI super-complex; T.S. and N.D. identified gene products of FCP and PSI core by MS analyses; K.I, I.U., and Y.K. provided genome information of *C. gracilis*; K.I. and M.K. performed phylogenic analysis; N.M. collected cryo-EM images; R.N. processed the EM data and reconstructed the final EM maps; R.N., K.K., N.M., and F.A. built the structure model; K.K. refined the final models; F.A. analyzed the structure; R.N. and S.A. proposed structural interpretation on the basis of spectroscopic analyses; R.N., K.K., K.I., J.-R.S., N.M., and F.A. wrote the paper, and all of the authors contributed to the interpretations of the results and improvement of the manuscript.

## Competing interests

The authors declare no competing interests.

## Additional information

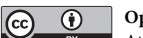

