## [Peer Review File · Nature Communications]

Reviewers' comments:

Reviewer #1 (Remarks to the Author):

The manuscript by Nagao et al. describes a cryo-EM structure of PSI-FCPI supercomplex from a centric diatom *C. gracilis* at a resolution of 2.4 Å. Diatoms are a major group of algae, greatly contributing to the primary production in the ocean. The structures of diatom PSII-FCPII were determined recently, and now the structural information of diatom PSI-FCPI is available, which is critical for understanding the subunit assembly and energy transfer processes of this supercomplex.

In this work, the authors solved the structure of diatom PSI-FCPI at a very high resolution. The structure shows that the diatom PSI core contains two additional subunits which are absent in the PSI core from other oxyphototrophs. The monomeric core associates with 16 FCPI monomers, which can be divided into two types, the inner FCP ring and the peripheral FCP. The structure reveals the organization of FCPI proteins and the arrangement of cofactors, provides important insights into the energy transfer pathways within the PSI-FCPI supercomplex.

This is a remarkable work and will be of great interest in the field of photosynthesis. I only have a few minor comments and suggestions.

1. General comment: The paper is well written, however, the manuscript might be a little hard to read for non-experts. The description of the structure seems somewhat dry. The presentation could be improved, and it will be of great help to make summarization when describe the structure (Please see points 3 & 4).

2. The differences of the core subunits between diatom PSI and PSI from other oxyphototrophs are interesting. The loss and gain of the transmembrane subunits in diatom PSI are likely to be related with the FCP association. In the structure, another unknown subunit was observed at the stromal side of PSI core, but the figures are not clear and the text does not give sufficient information about this subunit. Could it be involved in ferredoxin binding? I encourage the authors to add one or two more sentences to describe this subunit and the diatom PSI core structure.

3. I strongly suggest that the authors add one supplementary figure or table to summarize the binding sites of chlorophylls and carotenoids in different Fcp proteins. Although the supplementary figure 6 shows the pigment binding sites in each Fcp proteins, it would be better to give an overall view by summarizing these binding sites and indicating the conserved and non-conserved ones. For example, three pairs of dimeric chls were found in Fcpa8, are they conserved in all other Fcp proteins?

4. It may be useful that the authors compare the FCPIs with FCPIIs as well as Lhcrcs from red algae, both their apo-protein structure and the bound pigments. In green lineage, the LHCI and LHCII have highly similar structure, is it also the case in diatom? Moreover, a red chl pair 603-609 was observed in Lhcrcs and suggested to play important roles in both energy transfer and quenching functions (Pi, et al. PNAS, 2018). This pair is conserved in LHCI and LHCII from green lineage. I wonder if one of the three pairs of dimeric chls in diatom PSI corresponds to the red chl pair 603-609 in Lhcrc? Moreover, do those carotenoids which closely associated with the dimeric chl pairs in FCPs have counterparts in LHCs from green lineage, for example, the carotenoid in L1 or L2 site?

5. Line 318-326, the quenching process is induced by lumen acidification, which could be sensed by the Asp/Glu located at the thylakoid lumen. I don't think the Asp/Glu at the stromal side is highly related to the quenching events. Please modify this part of the text.

6. Line 279, thoughthrough

Reviewer #2 (Remarks to the Author):

The manuscript reports a 2.40-Å resolution structure of the diatom photosystem I (PSI)-fucoxanthin chlorophyll a/c binding proteins (FCP) (PSI-FCPI) supercomplex by cryo-electron microscopy. The structure reveals that the supercomplex is composed of 16 different FCPI subunits and a monomeric PSI core. The FCPI subunit show different protein structures with different pigment contents and binding sites, forming a complicated pigment-protein network. Clearly this novel structure provides unique insights into the light-harvesting strategy in diatom PSI-FCPI.

Other comments

1. The comparison between the diatom PSI-LHCI and the PSI-LHCI from *C. merolae* and *C. reinhardtii* could be improved and more specified. For *C. merolae* two types of PSI-LHCI have been described (Pi et al., 2018 and Antoshvili et al. 2019) having 3 and 5 LHCI (Pi et al. 2018). The two additional ones, absent from the PSI-LHCI crystal structure (Antoshvili et al. 2019) but present in the cryo-EM structure (Pi et al., 2018) are also found in the green alga PSI-LHCI cryo-EM structures (as referenced in the MS, see also Ozawa et al., 2018)) and are absent from the vascular plant structures. This should be clarified. The position of the subunit corresponding to LHCA9 (*C. reinhardtii*) seems to be occupied by Fcpa1 and is therefore conserved between green and red algal and diatom PSI-LHCI. The position of LHCA2 (*C. reinhardtii*) and correspondingly LHCr1 (*C. merolae*) is apparently not occupied by Fcpa polypeptides.
2. Recently the structure of a minimal photosystem I from the green alga *Dunaliella salina* has been published, this work should be discussed and referenced (Perez-Boerema A (2020) Nat Plants 6: 321-327)
3. What is the identity of the unknown protein 1 found in the structure? The authors performed mass spectrometric analyses of isolated diatom PSI-LHCI particles. From these data, is there a candidate?
4. Regarding energy quenching, do the diatom PSI-LHCI particles isolated under the conditions used for cryo-EM perform energy quenching?
5. The authors claim that expression of FCPs is highly dynamic. In this regard is it possible that LhcX like proteins associate and be involved in energy quenching?
6. Supplementary figure 3, LHCI subunits of red and green alga as well as of vascular plants PSI-LHCI should be named. It may help to depict and label PSI-LHCI of these organisms independent and in addition to the overlay.
7. Is Psa28 exclusively present in the PSI-LHCI, or can it associate with membranes independent of PSI-LHCI. BN-PAGE fractionation may help to address this question.

Responses to the comments of Reviewer #1

(Remarks to the Author are shown in black):

The manuscript by Nagao et al. describes a cryo-EM structure of PSI-FCPI supercomplex from a centric diatom *C. gracilis* at a resolution of 2.4 Å. Diatoms are a major group of algae, greatly contributing to the primary production in the ocean. The structures of diatom PSII-FCPII were determined recently, and now the structural information of diatom PSI-FCPI is available, which is critical for understanding the subunit assembly and energy transfer processes of this supercomplex.

In this work, the authors solved the structure of diatom PSI-FCPI at a very high resolution. The structure shows that the diatom PSI core contains two additional subunits which are absent in the PSI core from other oxyphototrophs. The monomeric core associates with 16 FCPI monomers, which can be divided into two types, the inner FCP ring and the peripheral FCP. The structure reveals the organization of FCPI proteins and the arrangement of cofactors, provides important insights into the energy transfer pathways within the PSI-FCPI supercomplex.

This is a remarkable work and will be of great interest in the field of photosynthesis. I only have a few minor comments and suggestions.

Comment 1:

1. General comment: The paper is well written, however, the manuscript might be a little hard to read for non-experts. The description of the structure seems somewhat dry. The presentation could be improved, and it will be of great help to make summarization when describe the structure (Please see points 3 & 4).

Author reply 1:

First of all, we thank the reviewer for his/her positive evaluation and important suggestions to improve our manuscript. Based on the comments of the Reviewer, we modified our manuscript, especially the description of the structure, to improve the readability of the paper. The changes we made are described below; thank you.

Comment 2:

2. The differences of the core subunits between diatom PSI and PSI from other oxyphototrophs are interesting. The loss and gain of the transmembrane subunits in diatom PSI are likely to be related with the FCP association. In the structure, another unknown subunit was observed at the stromal side of PSI core, but the figures are not clear and the text does not give sufficient information about this subunit. Could it be involved in ferredoxin binding? I encourage the authors to add one or two

more sentences to describe this subunit and the diatom PSI core structure.

Author reply 2:

The sequences of the novel stromal subunit, Unknown1, were not identified in this study; however, its structure is clearly different from that of ferredoxin (see attached Figure). Also, the binding site of Unknown1 is different from that of ferredoxin. Unknown1 is tightly associated with PsaD as shown in a new Figure as Supplementary Fig. 5, whereas ferredoxin was attached to a pocket formed by

PsaA, PsaC and PsaE as shown by Kubota-Kawai et al. (Nat. Plants, 2018). To explain this, we added several sentences into the main text (page 5) and the detailed structures of Psa28 and Unknown1, and the binding site of Unknown1 in comparison with that of ferredoxin into a new Supplementary Fig. 5 of the revised manuscript.

Comment 3:

3. I strongly suggest that the authors add one supplementary figure or table to summarize the binding sites of chlorophylls and carotenoids in different Fcp proteins. Although the supplementary figure 6 shows the pigment binding sites in each Fcp proteins, it would be better to give an overall view by summarizing these binding sites and indicating the conserved and non-conserved ones. For example, three pairs of dimeric chls were found in Fcpa8, are they conserved in all other Fcp proteins?

Author reply 3:

Based on the comments of the Reviewer, we summarized the number of Chls and Cars and the binding sites of Chls in each FCP subunit in a new Supplementary Table 4 in the revised manuscript. As can be seen from the Table, Chls a302/a306 and Chls c308/c309 are completely conserved among the 16 Fcpa subunits, whereas Chls c319/c321 exist only in Fcpa8. In addition, a dimeric pair of Chls a303/a313 is found only in Fcpa7 but not in the other 15 Fcpa subunits. Interestingly, the Chls a303/a313 pair is not formed in Fcpa2 and Fcpa6, albeit with the presence of both Chls a303 and a313. This is due to the different loop structure between Fcpa7 and Fcpa2/6. To explain this, we added a new Supplementary Fig. 11 and several sentences (page 7) to the revised manuscript.

Comment 4:

4. It may be useful that the authors compare the FCPIs with FCPIIs as well as Lhcrs from red algae, both their apo-protein structure and the bound pigments. In green lineage, the LHCIs and LHCIIIs

have highly similar structure, is it also the case in diatom? Moreover, a red chl pair 603-609 was observed in Lhcrs and suggested to play important roles in both energy transfer and quenching functions (Pi, et al. PNAS, 2018). This pair is conserved in LHCI and LHCII from green lineage. I wonder if one of the three pairs of dimeric chls in diatom PSI corresponds to the red chl pair 603-609 in Lhcr? Moreover, do those carotenoids which closely associated with the dimeric chl pairs in FCPs have counterparts in LHCs from green lineage, for example, the carotenoid in L1 or L2 site?

Author reply 4:

Based on the comments of the Reviewer, we compared the structures of Fcpa8 (FCPI) with FCPII, LHCR, LHCII, or LHCI, which were shown in new Supplementary figures, Supplemental Figs. 14-17. The RMSD of the FCPI with FCPII, LHCR, LHCII and LHCI are 1.83–2.21 Å. Most of the Chl and Car-binding sites in the LHCR are conserved in FCPI, whereas the binding sites in FCPII, LHCII and LHCI are similar to those in FCPI to a lesser extent. These observations suggest that the protein structure and pigment-binding sites are well conserved in the LHCI family of the red lineages. It is interesting to note that the dimeric Chl sites of Chls a302/a306 and Chls a308/c309 in FCPI are completely conserved in FCPII, LHCR, LHCII and LHCI, implying that oxyphototrophs may rely on the dimeric Chls for light-harvesting strategy including EET and quenching, irrespective of the red and green lineages. To explain this, we added a paragraph (page 12) and new Supplementary Figs. 14-17 to the revised manuscript.

Comment 5:

5. Line 318-326, the quenching process is induced by lumen acidification, which could be sensed by the Asp/Glu located at the thylakoid lumen. I don't think the Asp/Glu at the stromal side is highly related to the quenching events. Please modify this part of the text.

Author reply 5:

We agree with your comments, and removed the sentence “There are also Asp and Glu residues located in the vicinity of Chl-Car pairs of Chl a/Fx, Chl a/Ddx and Chl c/Fx at the stromal side, which can be seen in all of Fcpa1–16 subunits.” (page 11) from the revised manuscript.

Comment 6:

6. Line 279, thoughthrough

Author reply 6:

We modified it; thank you.

Responses to the comments of Reviewer #2

(Remarks to the Author are shown in black):

The manuscript reports a 2.40-Å resolution structure of the diatom photosystem I (PSI)-fucoxanthin chlorophyll a/c binding proteins (FCP) (PSI-FCPI) supercomplex by cryo-electron microscopy. The structure reveals that the supercomplex is composed of 16 different FCPI subunits and a monomeric PSI core. The FCPI subunit show different protein structures with different pigment contents and binding sites, forming a complicated pigment-protein network. Clearly this novel structure provides unique insights into the light-harvesting strategy in diatom PSI-FCPI.

Other comments

Comment 1:

1. The comparison between the diatom PSI-LHCI and the PSI-LHCI from *C. merolae* and *C. reinhardtii* could be improved and more specified. For *C. merolae* two types of PSI-LHCI have been described (Pi et al., 2018 and Antoshvili et al. 2019) having 3 and 5 LHCI (Pi et al. 2018). The two additional ones, absent from the PSI-LHCI crystal structure (Antoshvili et al. 2019) but present in the cryo-EM structure (Pi et al., 2018) are also found in the green alga PSI-LHCI cryo-EM structures (as referenced in the MS, see also Ozawa et al., 2018)) and are absent from the vascular plant structures. This should be clarified. The position of the subunit corresponding to LHCA9 (*C. reinhardtii*) seems to be occupied by Fcpa1 and is therefore conserved between green and red algal and diatom PSI-LHCI. The position of LHCA2 (*C. reinhardtii*) and correspondingly LHCr1 (*C. merolae*) is apparently not occupied by Fcpa polypeptides.

Author reply 1:

First of all, we thank the reviewer for his/her valuable comments to improve our manuscript. The information of the red algal PSI-LHCI by X-ray crystallography was added to Introduction (page 3) and Discussion (page 11) sections. In particular, we modified and improved the structural comparisons between the diatom PSI-FCPI and the PSI-LHCI from *C. merolae* and green algae including *Dunaliella* in the Discussion section (page 11).

Comment 2:

2. Recently the structure of a minimal photosystem I from the green alga *Dunaliella salina* has been published, this work should be discussed and referenced (Perez-Boerema A (2020) Nat Plants 6: 321-327)

Author reply 2:

As described in Author reply 1, we cited this reference and added explanations about the structure

of *Dunaliella salina* to the revised manuscript.

Comment 3:

3. What is the identity of the unknown protein 1 found in the structure? The authors performed mass spectrometric analyses of isolated diatom PSI-LHCI particles. From these data, is there a candidate?

Author reply 3:

We performed MS analysis of the diatom PSI-FCPI, and carefully checked the MS data. However, Unknown1 could not be identified. So, there is no information about the candidate of Unknown1 at present.

Comment 4:

4. Regarding energy quenching, do the diatom PSI-LHCI particles isolated under the conditions used for cryo-EM perform energy quenching?

Author reply 4:

The quenching capacity seems to be conserved in the isolated PSI-FCPI particles used for cryo-EM analysis, because the number of FCPI subunits in the present cryo-EM PSI-FCPI particles is virtually identical to that in the PSI-FCPI membranes that were employed for TRF spectroscopy. As described in Nagao et al. (Photosynth. Res., 2019, 140:141-149), the present cryo-EM PSI-FCPI particles were prepared by sucrose density gradient centrifugation (SDGC) after solubilization of the PSI-FCPI membranes with DDM. The SDGC profile showed almost no dissociation of FCPI subunits. From these observations, the energy-quenching ability observed in the PSI-FCPI membranes is likely preserved in the isolated PSI-FCPI particles used for cryo-EM analysis.

Comment 5:

5. The authors claim that expression of FCPs is highly dynamic. In this regard is it possible that Lhcx like proteins associate and be involved in energy quenching?

Author reply 5:

There are three Lhcx genes in the genome of *C. gracilis*. In our preliminary data, deletion mutants of the Lhcx genes in *C. gracilis* showed lower quenching capability compared with the wild-type cells, indicating that the Lhcx family in this diatom contributes to the energy quenching. However, the Lhcx proteins were not found in the structure of the PSI-FCPI complexes, implying that the Lhcx proteins are not related to the quenching events in the PSI-FCPI. In any cases, further experiments are needed to address this issue.

Comment 6:

6. Supplementary figure 3, LHCI subunits of red and green alga as well as of vascular plants PSI-LHCI should be named. It may help to depict and label PSI-LHCI of these organisms independent and in addition to the overlay.

Author reply 6:

According to your comments, we added the name of LHCI in the PSI-LHCI structures of red alga, green alga and higher plant to Supplementary Fig. 3, and improved the discussions regarding them in the Discussion section (pages 11-12).

Comment 7:

7. Is Psa28 exclusively present in the PSI-LHCI, or can it associate with membranes independent of PSI-LHCI. BN-PAGE fractionation may help to address this question.

Author reply 7:

Since Psa28 is a trans-membrane subunit found in the structure of the isolated PSI-FCPI, we believe that it is also associated with the membrane. From the structure, it is located in the interface between the PSI core and the inner FCP ring, and even bind one Chl that may function to mediate the energy transfer from FCP to the PSI core. Thus, we consider that it is an intrinsic component of the diatom PSI core. To explain this more clearly, we added a sentence about Psa28 (page 5) to the revised manuscript and the detailed position and structure of Psa28 into the new Supplementary Fig. 5.